# MVDRAG3D: DRAG-BASED CREATIVE 3D EDITING VIA MULTI-VIEW GENERATION-RECONSTRUCTION PRIORS

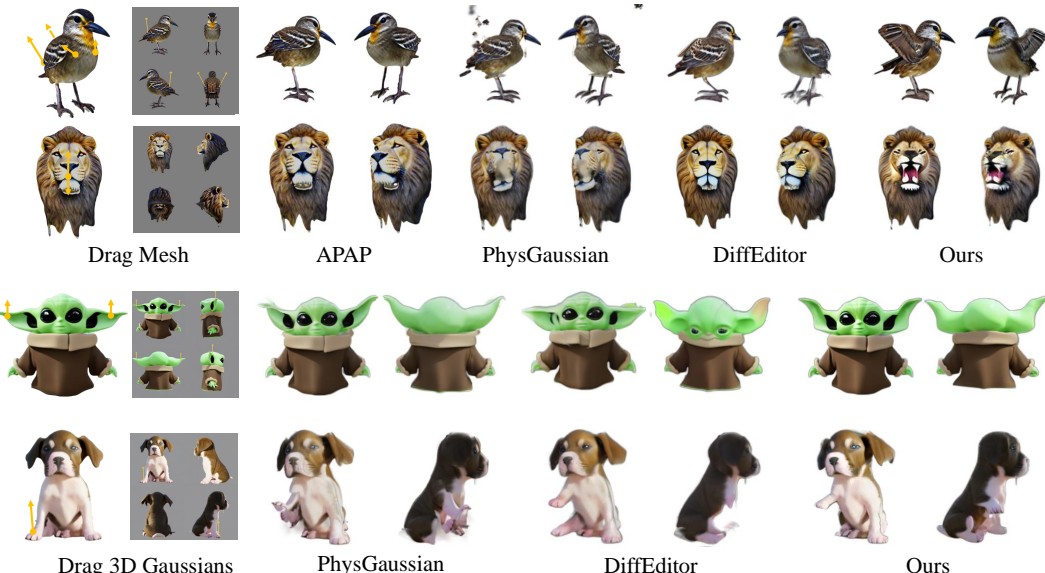

Figure 1: Comparison of our MVDrag3D with state-of-the-art approaches. The first two rows present results of dragging on meshes, while the last two focus on 3D Gaussians. Notably, APAP (Yoo et al., 2024) is specifically designed for mesh structures, and thus, it was not tested on 3D Gaussians. Overall, our method demonstrates the ability to produce more plausible and generative editing results, showing better performance across both 3D Gaussians and meshes.

## ABSTRACT

Drag-based editing has become popular in 2D content creation, driven by the capabilities of image generative models. However, extending this technique to 3D remains a challenge. Existing 3D drag-based editing methods, whether employing explicit spatial transformations or relying on implicit latent optimization within limited-capacity 3D generative models, fall short in handling significant topology changes or generating new textures across diverse object categories. To overcome these limitations, we introduce **MVDrag3D**, a novel framework for more flexible and creative drag-based 3D editing that leverages multi-view generation and reconstruction priors. At the core of our approach is the usage of a multi-view diffusion model as a strong generative prior to perform consistent drag editing over multiple rendered views, which is followed by a reconstruction model that reconstructs 3D Gaussians of the edited object. While the initial 3D Gaussians may suffer from misalignment between different views, we address this via view-specific deformation networks that adjust the position of Gaussians to be well aligned. In addition, we propose a multi-view score function that distills generative priors from multiple views to further enhance the view consistency and visual quality. Extensive experiments demonstrate that MVDrag3D provides a precise,

generative, and flexible solution for 3D drag-based editing, supporting more versatile editing effects across various object categories and 3D representations.

# 1 INTRODUCTION

Deforming 3D shapes by dragging point handles has been an essential interactive tool in computer graphics, enabling intuitive manipulation of complex shapes and structures. Traditionally, such drag-based 3D editing is often defined on mesh structures, utilizing optimization functions to preserve specific properties under the constraint of control handles. These properties include the mesh Laplacian (Lipman et al., 2004; 2005; Sorkine et al., 2004), local rigidity (Igarashi et al., 2005; Sorkine & Alexa, 2007), and surface Jacobians (Aigerman et al., 2022; Gao et al., 2023), as well as more recent considerations of perceptual plausibility (Yoo et al., 2024). However, these methods are constrained by the fixed topology of mesh structures, limiting their flexibility, especially in complex edits that require substantial changes to the topology or the generation of new textures, e.g., editing a bird to open its wings.

In light of the recently introduced 3D Gaussian splatting (Kerbl et al., 2023) that is more expressive and easy to edit, Interactive3D (Dong et al., 2024) introduces a series of deformable and rigid 3D operations to directly manipulate local 3D Gaussians. This is followed by Gaussian-to-NeRF reformatting and refinement through Score Distillation Sampling (SDS) (Poole et al., 2022). However, this method suffers from prolonged NeRF optimization and the typical limitations of vanilla SDS, such as over-saturation. PhysGaussian (Xie et al., 2024) also simulates drag-induced motion by integrating physically grounded dynamics into 3D Gaussians. However, it requires an accurate predefinition of the physical properties involved, which can be difficult to obtain. Besides, both methods still face challenges in making large structural changes and generating new content.

Notably, recent drag-based editing has seen considerable success in the 2D domain (Pan et al., 2023; Mou et al., 2023; 2024; Zhang et al., 2024; Shin et al., 2024), largely due to the capabilities of powerful image generative models, such as GANs (Karras et al., 2020) and diffusion models (Rombach et al., 2022). These models encompass a latent space that enables various harmonious manipulations, including object deformation, layout adjustments, and coherent new content generation. Building on this success, some 3D editing methods have begun to explore generative 3D dragging within a 3D latent space. For instance, Drag3D (Tang, 2023), adapts DragGAN (Pan et al., 2023) by incorporating a 3D GAN (Shen et al., 2021) into a motion-based latent optimization framework. Similarly, CNS-Edit (Hu et al., 2024) employs a latent-based method but combines it with a 3D neural volume diffusion model (Hui et al., 2022). This approach requires training separate models for each shape category, making it less flexible and more resource-intensive. Obviously, both of the above approaches are limited by the capacity and generalization of current 3D generative models.

In pursuit of a stronger generative prior for more powerful drag-based 3D editing, we have observed the following from existing 3D generation and reconstruction work: 1) most 3D representations can be rendered into multiple views; 2) 3D objects can be faithfully reconstructed from four and more views (Tang et al., 2024a; Xu et al., 2024b); and 3) existing multi-view diffusion models provide a strong prior for generating consistent images across four orthogonal views (Shi et al., 2023b; Kant et al., 2024). These observations inspire us to explore the potential of leveraging both *large-scale multi-view generation and reconstruction models* as 3D priors, agnostic to 3D representations, to facilitate precise, generative, and general 3D dragging. Ideally, we expect that the 3D dragging operation should exhibit the following properties 1) *Accuracy*: the ability to precisely drag any point on a 3D object's surface to a target spatial position; 2) *Generative capability*: the ability to generate visually plausible new content to match the drag intention; and 3) *Versatility*: compatibility with various input object categories and most 3D representations, such as 3D Gaussians or meshes.

To this end, we introduce MVDrag3D, a novel framework for drag-based 3D editing that leverages multi-view generation and reconstruction priors. Our method begins by rendering four orthogonal views of a 3D object and projecting the dragging points onto the corresponding views. To ensure consistent 3D edits, we extend the score-based gradient guidance mechanism within a multi-view diffusion model and propose a multi-view guidance energy function, enabling consistent edits across all four views. Thanks to the generative capabilities of the multi-view diffusion model, edits across four views can faithfully reflect significant structural changes or newly synthesized textures. The edited views are then fused into a 3D Gaussian representation using a multi-view Gaussian recon-

struction model. Although the initial 3D Gaussian appears complete, we observe a loss of appearance detail, and the 3D Gaussians in the overlapping regions between views do not align accurately, leading to noticeable discrepancies in the 2D rendering. To address these issues, we employ a deformation network that predicts the displacement of each Gaussian to correct the 3D alignment. Additionally, we formulate an image-conditioned multi-view score function to distill generative priors from the multiple views simultaneously, ensuring high-fidelity results while preserving details across all views. We summarize our contributions as follows:

1. We propose MVDrag3D, a drag-based 3D editing framework that leverages multi-view generation-reconstruction priors. It is accurate, generative, and adaptable to diverse input categories and most 3D representations, such as 3D Gaussians and meshes.

2. We extend the gradient guidance mechanism into a multi-view diffusion model and introduce multi-view guidance energy, which ensures consistent drag-based edits across four views.

3. We design a lightweight deformation network that corrects each 3D Gaussian's position and enhances geometric consistency. Furthermore, we introduce an image-conditioned multi-view score function to iteratively refine the 3D Gaussian, ensuring high-fidelity appearance and preserving fine details across all views.

## 2 RELATED WORK

We will review prior research, starting from drag-based 2D image editing techniques, and progressing to more recent developments in drag-based 3D editing and 3D generation-reconstruction priors.

**Drag-based image editing**. Drag-based image manipulation allows users to exert precise control over specific areas of the image via manual interactions like dragging and clicking. Most existing techniques employ iterative latent optimization in the latent space, and they can be roughly divided into two categories: methods that rely on motion tracking (Pan et al., 2023; Shi et al., 2024; Zhang et al., 2024; Cui et al., 2024; Liu et al., 2024a; Ling et al., 2024) and those based on guidance gradients (Mou et al., 2023; 2024). DragGAN (Pan et al., 2023), for instance, optimizes the latent space of GANs using iterative motion supervision and point tracking. Later, diffusion-based methods, including DragDiffusion (Shi et al., 2024), GoodDrag (Zhang et al., 2024), StableDrag (Cui et al., 2024), DragNoise (Liu et al., 2024a), and FreeDrag (Ling et al., 2024), have further refined these motion-driven techniques for more refined results. Meanwhile, DragonDiffusion (Mou et al., 2023) and DiffEditor (Mou et al., 2024) utilize a gradient-based approach by optimizing an energy function (Epstein et al., 2023) to achieve desired edits. Since both motion- and gradient-based methods require time-consuming iterations, SDEDrag (Nie et al., 2024) and FastDrag (Zhao et al., 2024) have been proposed to accelerate the editing process. More recently, InstantDrag (Shin et al., 2024) decomposes the dragging task into two components: learning motion dynamics and generating images conditioned on motion, achieving a better balance among interactivity, speed, and quality.

**Drag-based 3D editing**. To achieve drag-based 3D editing, classical mesh deformation techniques are commonly employed. These methods often design optimization functions to preserve specific geometric properties, such as the mesh Laplacian (Lipman et al., 2004; 2005; Sorkine et al., 2004), local rigidity (Igarashi et al., 2005; Sorkine & Alexa, 2007), and surface Jacobians (Aigerman et al., 2022; Gao et al., 2023), under the constraints of user-interactive handles like key points or cages. Despite their widespread use, these techniques frequently result in unnatural shape distortion, primarily due to their inability to ensure perceptual plausibility. To address this limitation, APAP (Yoo et al., 2024) introduced an innovative approach by incorporating SDS loss to optimize the Jacobian deformation field. However, like previous mesh deformation methods, APAP is constrained by the fixed topology of mesh structures, limiting its flexibility, particularly for complex edits that require generating entirely new content. On the other hand, Interactive3D (Dong et al., 2024) introduces a series of deformable and rigid 3D point operations on 3D Gaussians and also employs SDS to optimize the deformed or transformed Gaussians/NeRFs. Besides, PhysGaussian (Xie et al., 2024) also involves certain types of drag-related motion by integrating physically grounded dynamics into 3D Gaussians, however, it requires a suitable predefinition of the physics involved. Although these latter two methods employ more expressive 3D representations, they often require labor-intensive post-processing and face challenges in refining fine details or generating coherent new content.

As drag-based image editing techniques evolve, some 3D editing methods have begun to explore generative 3D dragging within a 3D latent space. For instance, Drag3D (Tang, 2023), built upon DragGAN (Pan et al., 2023), integrates a 3D GAN model into a motion-based latent optimization framework. However, the approach is inherently limited by the capacity and generalization constraints of current 3D GAN models. Later, CNS-Edit (Hu et al., 2024) introduces a coupled neural shape representation to facilitate 3D shape editing. This method utilizes a latent code to capture high-level global semantics, while a 3D neural feature volume provides spatial context for local shape modifications. However, CNS-Edit's category-specific design requires separate models for different 3D shape categories. Different from them, in this work, we achieve 3D generative dragging within a more powerful multi-view latent space.

**Multi-view Image Generation**. 2D diffusion models (Rombach et al., 2022; Saharia et al., 2022) initially focus on generating a single-view image. Recently, several models (Shi et al., 2023b; Wang & Shi, 2023; Shi et al., 2023a; Li et al., 2023b; Long et al., 2024; Kant et al., 2024; Tang et al., 2024b; Liu et al., 2024b) turned to employ a 3D-aware multi-view diffusion approach, incorporating camera poses as additional inputs and fine-tuning the diffusion model on multi-view data (Deitke et al., 2023). This strategy enables the consistent generation of multi-view images representing the same object. Essentially, these multi-view diffusion models capture a rich, generalizable distribution of 3D data, agnostic to a specific 3D representation. Also, given the limitations of current "pure" 3D generative models—those trained directly on 3D data—we believe that leveraging multi-view diffusion models as a 3D prior proxy could offer a promising solution for flexible 3D editing.

**Feed-forward Multi-view 3D Reconstruction**. By generating 3D-consistent multi-view images, various optimization techniques can be employed to reconstruct 3D objects (Shi et al., 2023b; Wang & Shi, 2023; Liu et al., 2023). To improve generation speed and quality, more recent work has explored large-scale reconstruction models using multi-view images (e.g., 4 or 6) (Wang et al., 2023; Xu et al., 2023; Li et al., 2023a; Wang et al., 2024; Xu et al., 2024a). These approaches leverage transformers to directly regress triplane-based NeRF representations. Newer methods like LGM (Tang et al., 2024a) and GRM (Xu et al., 2024b) replaced triplane NeRF with 3D Gaussians (Kerbl et al., 2023), achieving high-fidelity rendering at faster speeds. In summary, these recent feed-forward multi-view reconstruction models provide a robust 3D reconstruction prior, enabling the fast and faithful recreation of complete 3D objects from sparse-view images. In this work, we utilized a 4-view reconstruction model (Tang et al., 2024a) and a 4-view diffusion model (Shi et al., 2023b) as our generation-reconstruction priors.

## 3 METHOD

In this section, we briefly introduce score-based guidance energy for image editing, followed by a detailed explanation of our method.

### 3.1 PRELIMINARY

**Score-based gradient guidance for image editing.** Recently, DragonDiffusion (Mou et al., 2023) and DiffEditor (Mou et al., 2024) have applied score-based gradient guidance (Dhariwal & Nichol, 2021) to efficient and flexible image-editing tasks. The score function enables sampling from a more enriched distribution, generally defined as:

$$\tilde{\boldsymbol{\epsilon}}_\theta^t(\mathbf{x}_t) = \boldsymbol{\epsilon}_\theta^t(\mathbf{x}_t) + \eta \cdot \nabla_{\mathbf{x}_t} \mathcal{E}(\mathbf{x}_t, \mathbf{y}), \tag{1}$$

where the first term is the unconditional denoiser, and the second term is the conditional gradient produced by an energy function. Here, $\eta$ is the learning rate, and $\mathbf{y}$ represents the edit target, such as text embedding. During the diffusion sampling process, the gradient guidance from the energy function aligns with the editing target, gradually modifying the input image to meet the desired edit.

In recent 2D dragging task (Mou et al., 2024; 2023), the guidance energy function is constructed based on image feature correspondence within a pre-trained diffusion model as follows:

$$\nabla_{\mathbf{z}_t} \log q(\mathbf{y}|\mathbf{z}_t) = \alpha \cdot \mathbf{m}_{edit} \cdot \nabla_{\mathbf{x}_t} \mathcal{E}_{edit} + \beta \cdot (1 - \mathbf{m}_{edit}) \cdot \nabla_{\mathbf{x}_t} \mathcal{E}_{content}, \tag{2}$$

where $\mathbf{m}_{edit}$ is the editing region mask. The energy function $\mathcal{E}_{edit}$ measures the diffusion feature similarity between areas near the dragging start and destination points, while $\mathcal{E}_{content}$ ensures that

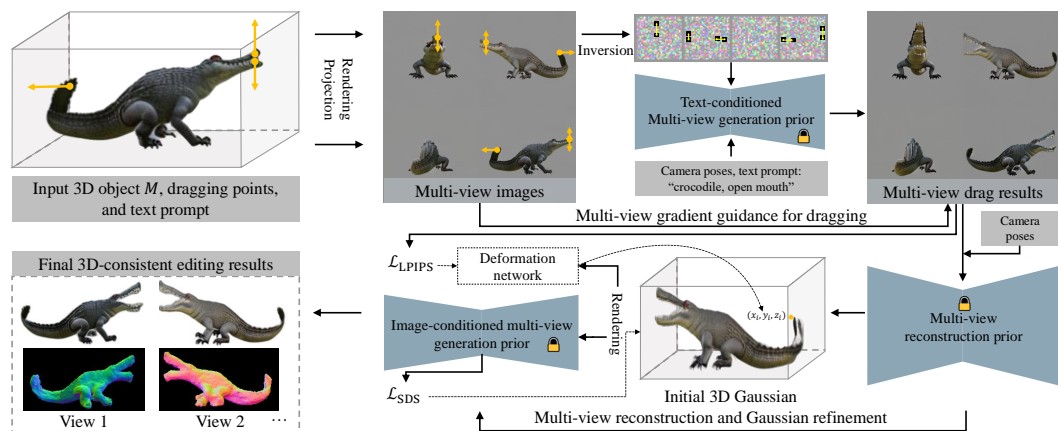

Figure 2: Method overview. Given a 3D model and multiple pairs of 3D dragging points, we first render the model into four orthogonal views, each with corresponding projected dragging points. Then, to ensure consistent dragging across these views, we define a multi-view guidance energy within a multi-view diffusion model. The resulting dragged images are used to regress an initial set of 3D Gaussians. Our method further employs a two-stage optimization process: first, a deformation network adjusts the positions of the Gaussians for improved geometric alignment, followed by image-conditioned multi-view score distillation to enhance the visual quality of the final output.

unedited content stays consistent with the original image. $\alpha$ and $\beta$ are balance weights. In our work, we extend both the editing energy and content energy to a multi-view version. This ensures that modifications made in one view are coherently reflected across all views.

## 3.2 OVERVIEW

The entire process is visualized in Fig. 2. Given a 3D model $M$ to be edited, and $k$ pairs of 3D dragging points $\{(\mathbf{p}_j^{3D}, \mathbf{q}_j^{3D})\}_{j=1}^k$, we first render $M$ into four orthogonal images $\mathcal{I} = \{\mathbf{I}_i\}_{i=1}^4$, along with the corresponding dragging points (Sec. 3.3). We then propose a multi-view guidance energy function (Sec. 3.4), which ensures consistent and coherent dragging across all views. The edited images $\mathcal{I}_e = \{\mathbf{I}_{e,i}\}_{i=1}^4$ are used to regress 3D Gaussians using (Tang et al., 2024a). While the initial reconstruction appears complete, we further use a deformation network and introduce an image-conditioned multi-view score distillation to correct the misalignment between Gaussians in the overlapping regions of each view and enhance the visual appearance across all views, resulting in the final edited results (represented in 3D Gaussians) (Sec. 3.5).

## 3.3 3D-2D RENDERING AND PROJECTION

We decompose the 3D dragging operation in a multi-view manner. First, we render the 3D model $M$ into four orthogonal images $\{\mathbf{I}_i\}_{i=1}^4$ using any suitable renderer. Since MVDream typically generates images with gray backgrounds, we adopt a similar gray background for rendering. In terms of camera setup, we adopt the same configuration as MVDream (Shi et al., 2023b) and LGM (Tang et al., 2024a), which serve as our generation-reconstruction priors. Specifically, the four views are chosen at orthogonal azimuths $(0°, 90°, 180°, 270°)$ and a fixed elevation $(0°)$. Then, the $k$ pairs of 3D dragging points can be projected onto the corresponding views, represented as $\{(\mathbf{p}_{i,j}^{2D}, \mathbf{q}_{i,j}^{2D})\}_{j=1}^k$. However, due to potential occlusions in certain views, we discard the point pairs if the $z$-axis value of $\mathbf{p}_{i,j}^{2D}$ or $\mathbf{q}_{i,j}^{2D}$ exceeds the rendered depth at the corresponding 2D position.

## 3.4 MULTI-VIEW GRADIENT GUIDANCE FOR DRAGGING

Since a 3D object can be rendered into multiple images and numerous drag-based 2D editing methods already exist, a straightforward approach to achieve drag-based 3D editing would be to independently edit each view and then reconstruct the 3D model. However, this leads to significant 3D inconsistencies (see the results of DiffEditor (Mou et al., 2024) in Fig. 1), as the editing results of

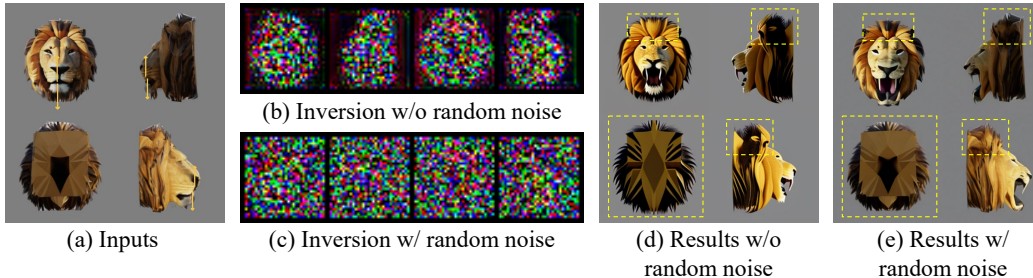

(a) Inputs     (b) Inversion w/o random noise / (c) Inversion w/ random noise     (d) Results w/o random noise     (e) Results w/ random noise

Figure 3: Effect of DDIM inversion with random noise. For the rendered four images, when inverted into MVDream's data distribution, the resulting noise deviates from a Gaussian distribution (b). By adding random noise ($\mathcal{N}(0, 0.01)$) to the background's pixel domain, we help the latent variables conform more closely to a Gaussian distribution (c). The resulting multi-view edits are shown in (d) and (e). Yellow dashed boxes indicate the regions with evident differences.

each image become misaligned across various factors such as pose, layout, texture, and more. Based on the observation that multi-view diffusion models can simultaneously generate a consistent set of multi-view images, and recognizing the effectiveness of score-based gradient guidance in image editing, we extend gradient guidance to a multi-view version.

Specifically, we first apply DDIM inversion (Song et al., 2020) to transform each of $\{\mathbf{I}_i\}_{i=1}^4$ into a Gaussian distribution. These distributions are combined and represented as $\mathbf{z}_T \in \mathcal{R}^{4 \times H \times W \times C}$ within the latent space of MVDream. Using $\mathbf{z}_T$, we can extract an intermediate feature $\mathbf{F}$ from the UNet decoder. Note that MVDream reshapes $\mathbf{z}_T$ into a $4HW \times C$ format, thus extending self-attention to the cross-view version. This ensures that guidance from one view can influence the others. With this, we follow (Mou et al., 2023) and define a multi-view guidance energy:

$$\mathcal{E}_{edit} = \sum_{i=1}^4 \frac{1}{0.5 \cdot \cos\left(\mathbf{F}_{i,t}^{edi}[\mathbf{m}_i^{edi}],\ sg(\mathbf{F}_{i,t}^{ori}[\mathbf{m}_i^{ori}])\right) + 0.5},$$

$$\mathcal{E}_{content} = \sum_{i=1}^4 \frac{1}{0.5 \cdot \cos\left(\mathbf{F}_{i,t}^{edi}[\mathbf{m}_i^{unedited}],\ sg(\mathbf{F}_{i,t}^{ori}[\mathbf{m}_i^{unedited}])\right) + 0.5},$$

$$(3)$$

where $\mathbf{F}_{i,t}^{edi}$ and $\mathbf{F}_{i,t}^{ori}$ are intermediate features of $\mathbf{z}_{i,t}^{edi}$ and $\mathbf{z}_{i,t}^{ori}$. $\mathbf{z}_{i,t}^{ori}$ corresponds to the latent variables of original image at time step $t$, while $\mathbf{z}_{i,t}^{edi}$ represents the edited latent variable. $sg(\cdot)$ is the gradient clipping operation. In the dragging operation, $\mathbf{m}^{ori}$ (or $\mathbf{m}^{edi}$) is a $3 \times 3$ rectangular patch centered around the 2D dragging points $\mathbf{p}^{2D}$ (or $\mathbf{q}^{2D}$). $\mathbf{m}^{unedited}$ denotes the areas without editing. To enhance readability, the index labels on each image are omitted. Note also that all layers of the UNet decoder features are used to compute the guidance energy, ensuring more comprehensive and robust results. The gradient of $\mathcal{E}_{edit}$ is then used to generate consistently edited images $\{\mathbf{I}_{e,i}\}_{i=1}^4$, while $\mathcal{E}_{content}$ employed to preserve the appearance of the unedited regions, keeping them as close to the original images as possible.

**DDIM inversion with random noise**. During DDIM inversion, we observed that for the given four images, their latent noise does not follow a Gaussian distribution, as depicted in Fig. 3 (b). This discrepancy often causes instability during the editing process, making it difficult to preserve the object's identity (see Fig. 3 (d)). We believe this issue arises because MVDream was never trained on images with smooth, noise-free regions like the background, leading to a domain gap during inversion (Ouyang et al., 2024). To address this issue, we found that introducing small, nearly imperceptible perturbations to the pixel domain—especially in smooth areas like the background—significantly improves the inversion process. These subtle disturbances help the latent variables conform more closely to a Gaussian distribution (see Fig. 3 (c)). The final results exhibit smoother transitions and better overall fidelity in the edited images, as shown in Fig. 3 (e).

### 3.5 3D GAUSSIAN RECONSTRUCTION AND REFINEMENT

Once we obtain the four edited images, we employ LGM (Tang et al., 2024a) to regress a partial 3D Gaussians for each view and then fuse them into a unified 3D Gaussian representation. However, we encountered two significant challenges: (1) because we only use

four orthogonal views, the predicted Gaussians in the overlapping regions between views are usually not aligned correctly, resulting in noticeable discrepancies in the 2D rendering (see Fig. 4 (c)), and (2) the appearance details are frequently lost during LGM's regression process, reducing the visual fidelity of the final 3D reconstruction (see Fig. 5 (c)).

In our early tests, to address these issues, we applied vanilla SDS on the initial reconstruction, incorporating a multi-view reconstruction loss across the four views. However, these adjustments did not resolve the underlying issues. We attribute these challenges to the inherent ambiguity in the SDS and reconstruction losses. Specifically, it is difficult to directly optimize independent Gaussians consistently without regularization, and the losses do not effectively indicate when to adjust the position or when to densify or prune the Gaussians, resulting in suboptimal outcomes. To address these challenges, we propose a two-step approach: first, we adjust the Gaussian's position via deformation fields to achieve better geometric alignment and then focus on enhancing visual quality.

**Gaussian position optimization**. Considering that the geometric misalignment problem across views mainly involves low-frequency overall structural changes and the Gaussians

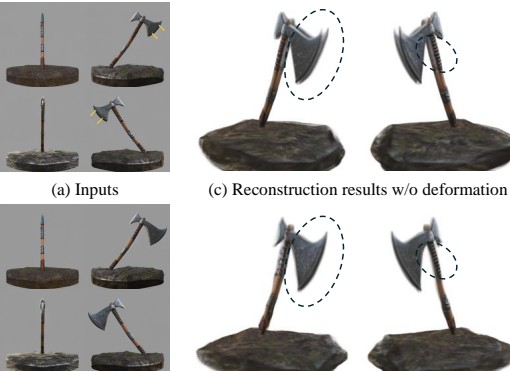

(a) Inputs     (c) Reconstruction results w/o deformation

(b) Multi-view drag results     (d) Reconstruction results w/ deformation

Figure 4: Effect of Gaussian position optimization. (c) shows 3D reconstruction result may exhibit structural misalignment. By employing a deformation network to optimize the Gaussian position, we achieve better compactness and consistency among the Gaussians across different views, as shown in (d).

belonging to the same view should be moved more consistently, for each view' Gaussian set, we propose to use an individual deformation network $f$ to predict each Gaussian's movement $(\delta x_i, \delta y_i, \delta z_i)$. This means we employ a total of four lightweight individual MLPs, one for each view. Besides, since standard MLPs are generally ineffective for low-dimensional coordinate-based regression tasks (Tancik et al., 2020), we enhance the model by applying Fourier positional embeddings $(pe(\cdot))$ to each Gaussian's $(x, y, z)$ coordinates. The new position for each Gaussian is then calculated as: $(x', y', z') = (x, y, z) + f(pe((x, y, z)))$. The training loss is the VGG-based LPIPS loss, applied to the four images. This helps maintain perceptual similarity and ensures better alignment across views: $\mathcal{L}_{\text{LPIPS}} = \sum_{i=1}^{4} \text{LPIPS}(\mathbf{I}_{e,i}, \mathbf{I}_{e,i}^{\text{render}})$, where $\mathbf{I}_{e,i}^{\text{render}}$ is the rendered image by the optimized Gaussians after their positions have been corrected. Note that Gaussian densification and pruning are not performed at this stage. Fig. 4 (d) shows the effectiveness of the Gaussian position optimization stage.

**Gaussian appearance optimization**. The deformation network described above is limited to optimize the positions of the Gaussians and is therefore unable to recover lost texture details during multi-view reconstruction. Drawing inspiration from ReconFusion (Wu et al., 2024a), we propose reframing the Gaussian appearance enhancement task as an image-conditioned multi-view SDS optimization problem. Our objectives are twofold: (1) to ensure multi-view consistency across novel camera angles beyond the initial four views, and (2) to preserve the identity of the edited four views. To achieve this, we define an edited-image-conditioned multi-view score function:

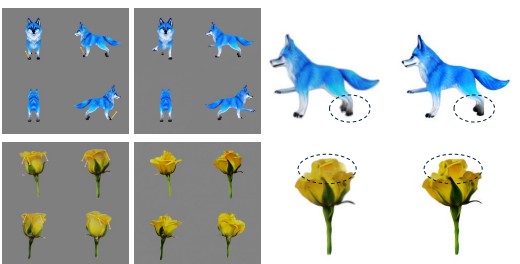

(a) Inputs   (b) Drag results   (c) w/o optimization   (d) w/ optimization

Figure 5: Effect of image-conditioned multi-view SDS. (c) presents the reconstruction results without appearance optimization, while (d) displays the corresponding results after optimization, which are sharper and clearer.

$$\nabla_\phi \mathcal{L}_{\text{SDS}} = \mathbb{E}_{t,\epsilon,o}[(\epsilon_\theta(\hat{I}; t, \mathbf{I}_{e,i}, o) - \epsilon)\frac{\partial \hat{I}}{\partial \phi}], \text{ and } i = 1, 2, 3, \text{ or } 4, \tag{4}$$

where $\hat{I}$ represents the rendered batch images from any four orthogonal views, and $o$ denotes the corresponding camera poses. During each SDS iteration, we randomly render four orthogonal views

and randomly select one edited image $\mathbf{I}_{e,i}$ as a condition to compute the SDS loss. The multi-view diffusion model employed is ImageDream (Wang & Shi, 2023), which can be seen as an image-conditioned version of MVDream. This allows it to be seamlessly integrated into our framework. In each iteration, we also compute $\mathcal{L}_{\text{LPIPS}}$. It is important to note that all Gaussian properties are optimized during this process. Additionally, following (Kerbl et al., 2023), we incorporate densification and pruning operations to create or remove Gaussians, to adjust inaccurately reconstructed regions.

# 4 EXPERIMENTS

## 4.1 EXPERIMENTAL SETUP

**Implementation Details**. We conducted all experiments on a single 48 GB A6000 GPU. For multi-view image dragging, we employed DDIM sampling with 150 steps, applying random Gaussian noise $\mathcal{N}(0, 0.01)$ to the background. In the Gaussian deformation stage, we used 4 MLPs, each trained for $2,000$ iterations with a learning rate of $0.00001$. Each MLP consists of a linear layer, a ReLU activation, and another linear layer arranged in a residual structure. For multi-view SDS optimization, we performed $1,000$ iterations, gradually decaying $T_{\max}$ from 0.49 to 0.02.

**Datasets**. We perform dragging on two of the most popular 3D representations: meshes and 3D Gaussians. For the mesh experiments, we collected 8 meshes from (Yoo et al., 2024) and *Genie* (Luma AI). For the 3D Gaussian experiments, we collected 8 3D Gaussians from Tang et al. (2024a). We collect data that are representative to demonstrate drag editing but do not cherry-pick based on any results. The 3D drag points are manually specified using MeshLab, following (Yoo et al., 2024).

**Metrics**. In this work, we employ two assessment metrics for quantitative evaluation: Dragging Accuracy Index (**DAI**) (Zhang et al., 2024) and **GPTEval3D** (Wu et al., 2024b). DAI measures the effectiveness of a method in transferring source content to a target point. While DAI effectively measures drag accuracy, it is insufficient because the editing process sometimes introduce overall distortions or artifacts, resulting in unrealistic or unnatural results. To address this, we use GPTEval3D, which leverages GPT-4V and customizable 3D-aware prompts to offer flexible comparisons between two 3D assets based on a set of specific evaluation criteria. For more details about these metrics, please refer to Sec. A.2.

## 4.2 RESULTS

**Baselines**. One baseline comparison involves leveraging a 2D drag method to edit each view independently. In this setup, we use DiffEditor (Mou et al., 2024) to drag the four rendered views, followed by the same reconstruction and optimization steps as ours to produce the final 3D results. During our initial experiments, we observed that when editing much more than four views, such as 120, DiffEditor introduced significant 2D inconsistencies. Thus, for a fair comparison, we limit the process to four images as in our approach. We also compare our method with APAP, the state-of-the-art drag-based mesh deformation technique. Additionally, we include PhysGaussian (Xie et al., 2024), which enables user control over Gaussian-based dynamics. For this comparison, we start with a 3D model, render four images, reconstruct a 3D Gaussian, and feed it into the PhysGaussian simulator. More detailed drag setup for PhysGaussian please refer to Sec. A.3. Note that as the released code of Interactive3D (Dong et al., 2024) cannot be run successfully, we are unable to include it in our comparisons. But conceptually, our approach provides a stronger multi-view diffusion prior compared to the SDS loss in Interactive3D, as we can also observe in our comparison with APAP.

**Visual Comparisons**. We first conduct a visual comparison of the proposed MVDrag3D against baselines, as demonstrated in Fig. 6. The first three rows present results of dragging on meshes, while the last three rows show results on 3D Gaussians. For each method, we render two views to highlight the respective editing results. Take the wolf mode in the first row as an example, we aim to lift its left leg. While APAP deforms the leg, it bends rather than lifts it, resulting in a less realistic motion. In contrast, our method produces an articulation-like motion that is more natural. DiffEditor generates a successful edit in some views, but others fail, leading to inconsistent 3D results. As for PhysGaussian, it relies on predefined physical properties. Since the optimal parameters are unknown, its results exhibit some distortion. Additionally, it is unable to generate new content. For more visual results, please refer to the supplemental video demo.

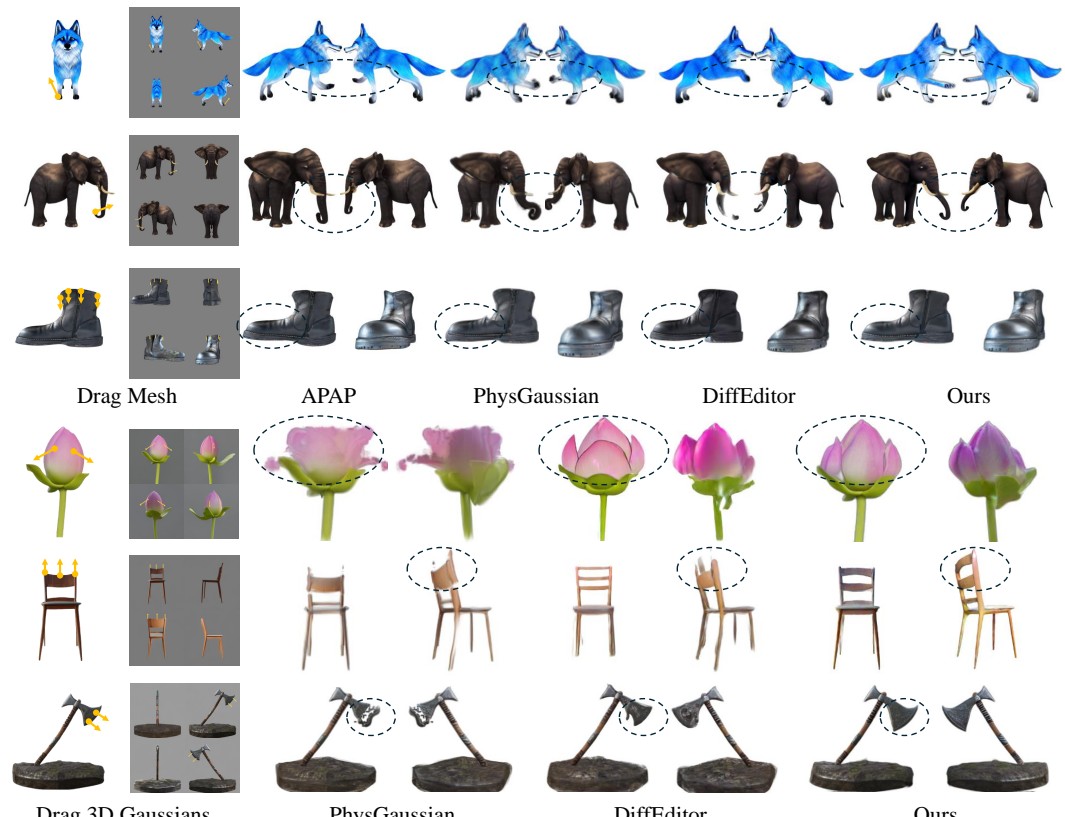

Figure 6: 3D dragging results on meshes and 3D Gaussians. The first three rows show the results for the mesh, and the last three rows show the results for the 3D Gaussians. Black dashed circles indicate some detailed differences.

Table 1: Quantitative comparison with state-of-the-art methods on both meshes and 3D Gaussians. Left side of "/": Mesh. Right side: 3D Gaussians. $\gamma$ represents the patch radius, which defines the neighborhood around the 2D dragging points. APAP was not tested on 3D Gaussians. In the last column, we report a rough average running time.

| Method | $\gamma = 1(\downarrow)$ | $\gamma = 3(\downarrow)$ | $\gamma = 5(\downarrow)$ | $\gamma = 7(\downarrow)$ | $\gamma = 10(\downarrow)$ | Time |
|---|---|---|---|---|---|---|
| APAP | 0.2154 / – | 0.2467 / – | 0.2150 / – | 0.1859 / – | 0.1672 / – | 6 minutes |
| PhysGaussian | 0.1763 / 0.2468 | 0.1887 / 0.2331 | 0.1671 / 0.2153 | 0.1448 / 0.1979 | 0.1296 / 0.1814 | 1 minutes |
| DiffEditor | 0.1564 / 0.1722 | 0.1452 / 0.1735 | 0.1348 / 0.1619 | 0.1299 / 0.1486 | 0.1300 / 0.1358 | 6 minutes |
| Ours (LGM) | 0.1153 / 0.1702 | 0.1080 / 0.1588 | 0.0989 / 0.1397 | **0.0890** / 0.1260 | **0.0865** / 0.1130 | 3 minutes |
| Ours + deformation | **0.1121** / 0.1269 | **0.1044** / 0.1150 | **0.0975** / 0.1081 | 0.0908 / 0.1017 | 0.0881 / 0.0937 | 5 minutes |
| Ours + deformation + SDS | 0.1461 / **0.1159** | 0.1292 / **0.1074** | 0.1175 / **0.1020** | 0.1064 / **0.0960** | 0.0994 / **0.0900** | 8 minutes |

**Quantitative Comparisons**. In addition to the visual comparisons, we conducted a quantitative evaluation to assess the effectiveness of all compared methods in terms of dragging accuracy (**DAI**) and overall editing quality (**GPTEval3D**). Table 1 reports different methods' DAI across varying patch radius values $\gamma$. As $\gamma$ increases from 1 to 10, our method, both with and without SDS, shows consistently lower error against other approaches like APAP, PhysGaussian, and DiffEditor. In Table 2, the GPTEval3D evaluation reveals that the "Ours + deformation + SDS" method performs almost the best across all criteria on both meshes and 3D Gaussians. Notably, we observed that while the SDS version of our method may not always achieve the highest DAI score, this is understandable. The SDS tends to sharpen visual details, which can lead to minor numerical decreases, but it ultimately results in more visually pleasing outputs. This is further supported by the GPTEval3D results, where the SDS version achieves the highest score in texture details.

Table 2: Evaluation results of GPTEval3D. "Ours + deformation + SDS" performs almost the best across all criteria on both meshes and 3D Gaussians.

| Method | Text-Asset Alignment (↑) | | 3D Plausibility (↑) | | Text-Geometry Alignment (↑) | | Texture Details (↑) | | Geometry Details (↑) | | Overall (↑) | |
|---|---|---|---|---|---|---|---|---|---|---|---|---|
| | Mesh | 3DGS | Mesh | 3DGS | Mesh | 3DGS | Mesh | 3DGS | Mesh | 3DGS | Mesh | 3DGS |
| APAP | 895.53 | – | 906.63 | – | 961.97 | – | 945.32 | – | 905.80 | – | 917.80 | – |
| PhysGaussian | 828.46 | 973.08 | 870.32 | 881.52 | 911.28 | 950.91 | 920.78 | 977.59 | 898.65 | 968.70 | 891.62 | 979.76 |
| DiffEditor | 982.32 | 883.25 | 1054.11 | 924.96 | 1045.48 | 868.99 | 1042.24 | 894.55 | 975.34 | 885.61 | 992.50 | 897.78 |
| Ours (LGM) | 1074.58 | 1047.74 | 1001.04 | 975.45 | **1090.78** | 1011.64 | 1075.72 | 959.59 | 1084.85 | 1026.61 | 1041.38 | 1048.89 |
| Ours + deformation | 1023.55 | 954.67 | 1060.81 | 947.32 | 1012.23 | 961.58 | 945.32 | 1066.18 | 1051.28 | 962.77 | 1066.18 | 982.10 |
| Ours + deformation + SDS | **1172.77** | **1113.36** | **1139.37** | **1103.98** | 1059.67 | **1122.44** | **1076.25** | **1098.33** | **1109.46** | **1108.64** | **1136.80** | **1100.33** |

Input         Multi-view dragging results         Input         Multi-view dragging results

Figure 7: Results of dragging on image-conditioned multi-view diffusion model. We extend the dragging stage to ImageDream (Wang & Shi, 2023). The results are less flexible as indicated by black arrows.

### 4.3 ABALATION AND DISCUSSION

**Abalation**. We start with the initial reconstruction from (Tang et al., 2024a) as a baseline (Ours (LGM)) and progressively integrate our two-step optimizations: (i) Gaussian position optimization (Ours + deformation), and (ii) image-conditioned multi-view SDS (Ours + deformation + SDS). Table 1 presents a clear comparison of the impact of each stage on both mesh data and 3D Gaussians. Fig. 4 and Fig. 5 also visually demonstrate the effectiveness of our proposed optimization strategy.

**Drag on image-conditioned diffusion model**. Considering the existence of several image-conditioned multi-view diffusion models, such as Imagedream (Wang & Shi, 2023) and Zero123++ (Shi et al., 2023a), an intuitive idea is to extend the multi-view dragging stage to these models. Here, we specifically extend it to Imagedream. Fig. 7 shows two cases. The conditioning image is the front view of each input. Under this setting, we observe that the results are less visually pleasing. We suspect the reason is that the image condition is too strong, thereby restricting the editing effects. In Mou et al. (2024), the authors introduce the use of both image and text for fine-grained image editing by tuning a new encoder, enabling a more detailed description of the desired changes. We see this as a potential direction for our work, aiming to enhance precision and flexibility in multi-view editing.

## 5 CONCLUSION

In this work, we introduce MVDrag3D, a novel paradigm that harnesses the power of multi-view generation-reconstruction priors for creative 3D editing. MVDrag3D first applies a multi-view dragging technique to ensure consistent edits across four orthogonal views. Following this, a reconstruction model generates 3D Gaussians of the edited object. To refine these initial 3D Gaussians, we introduce a deformation network that aligns the Gaussians across different views, complemented by a multi-view score function to enhance visual sharpness and consistency. Extensive experiments showcase the precision, generative capabilities, and flexibility of our method, making it a versatile solution for 3D editing across various object categories and representations.

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

# A APPENDIX

## A.1 ADDITIONAL PARAMETERS FOR MULTI-VIEW DRAGGING

For multi-view image dragging, parameters such as the editing and content energy balance weights $\alpha$ and $\beta$ (see Eq. 2) and the classifier-free guidance (CFG) need to be configured. We leave these as open parameters for users, as the optimal settings may vary depending on the specific edit target.

## A.2 METRIC EXPLANATION

**DAI**. DAI measures the effectiveness of a method in transferring semantic content to a target point. Specifically, it evaluates whether the content at the source position denoted as $p_j$, has been successfully moved to the target location $q_j$ in the edited 3D object. For each 3D object, the DAI is computed over four views and considers all non-occluded dragging points as follows:

$$\text{DAI} = \frac{1}{4} \sum_{i=1}^{4} \sum_{j=1}^{k} \frac{\left\| \mathbf{I}_i \cdot \Omega(\boldsymbol{p}_{i,j}^{2D}, \gamma) - \mathbf{I}_{e,i} \cdot \Omega(\boldsymbol{q}_{i,j}^{2D}, \gamma) \right\|_2^2}{(1 + 2\gamma)^2}, \tag{5}$$

where $\Omega(\boldsymbol{p}_{i,j}^{2D}, \gamma)$ represents a patch centered at $\boldsymbol{p}_{i,j}^{2D}$ with radius $\gamma$. Eq. 5 calculates the mean squared error between the patch at $\boldsymbol{p}_j^{2D}$ of $\mathbf{I}$ and the patch at $\boldsymbol{q}_j^{2D}$ of $\mathbf{I}_e$. By adjusting the radius $\gamma$, the metric can focus on different levels of context. A smaller $\gamma$ provides a precise evaluation of differences at the exact control points, while a larger $\gamma$ includes a broader region, allowing for an assessment of the surrounding context. This adaptability makes DAI a flexible tool for examining various aspects of editing quality. Given that the image resolution is $256 \times 256$, we set $\gamma = 1, 3, 5, 7, 10$.

**GPTEval3D**. While DAI effectively measures drag accuracy, it is not sufficient on its own because the editing process can introduce distortions or artifacts, leading to unrealistic or unnatural results. Therefore, evaluating the naturalness and fidelity of the edited images is crucial for a comprehensive quality assessment. This task is particularly challenging due to the absence of ground-truth edited 3D objects for reference. To address this, we utilize GPTEval3D, which leverages GPT-4V with customizable 3D-aware prompts. GPTEval3D aligns well with human judgment across several dimensions, including text-to-asset alignment, 3D plausibility, texture-–geometry coherence, texture details, and geometry details. Specifically, GPTEval3D prompts GPT-4V to compare two 3D assets generated by different methods using four rendered images and normal maps. The pairwise comparisons are then used to calculate Elo ratings, which reflect each method's performance. For more details, please refer to (Wu et al., 2024b).

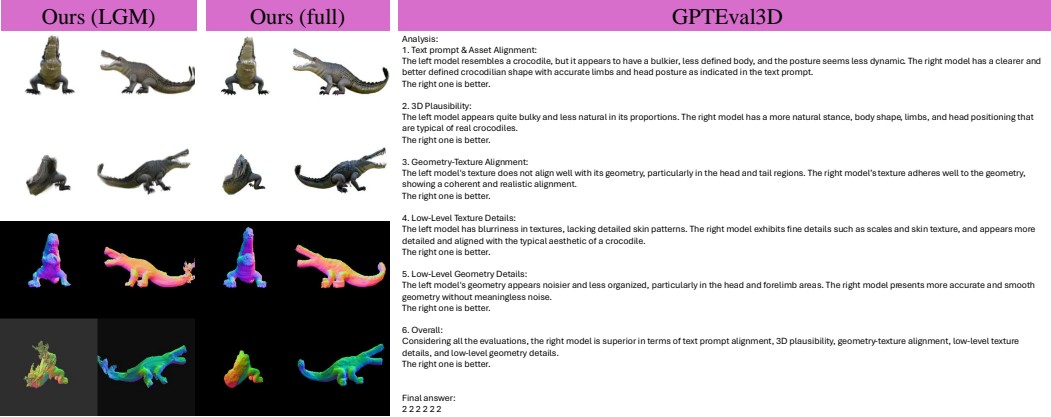

Figure 8: An analysis example of GPTEval3D on two versions of our method: Ours (LGM) and the full version, Ours + deformation + SDS. The left side of the figure shows selected four-view results from both methods, including both the appearance image and the normal map. On the right, GPT-4V's evaluation is presented, which aligns with human observations. The final line on the right confirms that the second method, Ours + deformation + SDS, outperforms the first, Ours (LGM), across all five evaluation criteria.

Fig. 8 presents a pairwise comparison example of GPTEval3D on two versions of our method: Ours (LGM) and the full version, Ours + deformation + SDS. The visual results on the left show that Ours (LGM) produces somewhat blurry output with noticeable noise in the geometry, particularly around the tail region. This can be attributed to the lack of optimization provided by the deformation network and SDS in this version. On the right side of the figure, GPT-4V's judgment aligns with our observations, concluding that the second method, Ours + deformation + SDS, outperforms Ours (LGM) across all five evaluation criteria.

### A.3 DRAG SETUP FOR PHYSGAUSSIAN

In PhysGaussian (Xie et al., 2024), we use the translation function as a proxy for the drag operation. We set the drag starting points as the center points and use the direction from the starting points to the destination points to define the initial velocity. For each dragging point pair, we assign a translation movement, and the simulation continues until either the starting point reaches the destination or the iteration count reaches the set maximum (75 by default).

### A.4 RUNNING TIME STATISTICS

The last column of Table 1 also summarizes the rough average running time for each method. APAP, DiffEditor, and the full version of our method are slower than PhysGaussian, Ours (LGM), and "Ours + deformation", mainly due to the absence of SDS optimization in their pipelines. PhysGaussian runs the fastest since it does not involve any optimization process.

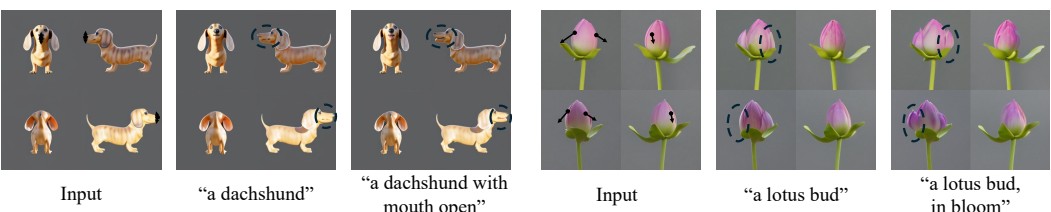

| Input | "a dachshund" | "a dachshund with mouth open" | Input | "a lotus bud" | "a lotus bud, in bloom" |

Figure 9: Effect of different text prompts. When editing images, a text prompt that better aligns with the drag intention can help query more meaningful features from the diffusion model, ultimately leading to more visually pleasing results. Black dashed circles highlight edit differences.

## A.5 TEXT PROMPT

Interestingly, during our early tests, we observed that text input plays a crucial cue for generative editing. As shown in Fig. 9, when dragging the dog's mouth to open, using a more specific text prompt like "a dachshund with an open mouth" can effectively guide the process. This proves the significance of prompt design in aligning the diffusion model's features with the intended edits. In all our experiments, we provide a more detailed text prompt when the drag intention is clear. However, for cases where the intention is less defined, we use a more general description instead.

## A.6 EFFECT OF DDIM INVERSION WITH RANDOM NOISE

Fig. 10 shows a new example to better illustrate the advantages of DDIM inversion with random noise. In this case, the editing intention is to lift the wolf's left leg, and the editing mask is applied solely to the area near the left leg, as shown in Fig. 10 (a). The regions outside the mask are expected to remain unchanged. However, as highlighted in the yellow dashed box in Fig. 10 (d), performing DDIM inversion without random noise leads to noticeable changes in the wolf's tail and many regions in the left-bottom view, even if these regions are outside of the editing mask. This occurs because the noise generated during DDIM inversion lacks precision and deviates from a Gaussian distribution, as shown in Fig. 10 (c). By introducing simple random noise processing, the DDIM inversion noise becomes more consistent with a Gaussian distribution, allowing regions outside the mask to better align with the original image.

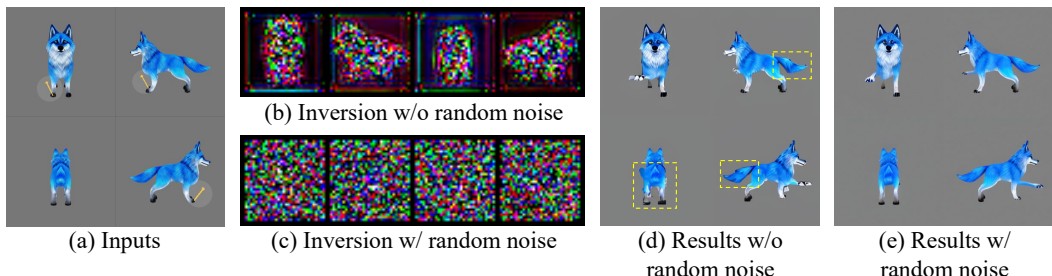

(a) Inputs     (c) Inversion w/ random noise     (d) Results w/o random noise     (e) Results w/ random noise

Figure 10: Effect of DDIM inversion with random noise. For the rendered four images, when inverted into MVDream's data distribution, the resulting noise deviates from a Gaussian distribution (b). By adding random noise ($\mathcal{N}(0, 0.01)$) to the background's pixel domain, we help the latent variables conform more closely to a Gaussian distribution (c). The resulting multi-view edits are shown in (d) and (e). Yellow dashed boxes indicate the regions with evident differences.

## A.7 SMOOTH SURFACE EXTRACTION

Since the final output of our method is a 3D Gaussians for 3D meshes, extracting a mesh model from the 3D GS may result in some loss of detail. Regarding improved mesh extraction, 2D GS could serve as a potential solution. Additionally, we came across the open-source work Lara (Chen et al., 2025), which uses four views to feed-forwardly regress a 2D GS model with a smoother surface. Fig. 11 shows the extracted mesh surface normal by Lara. In the future, we plan to release a Lara-based version of our method.

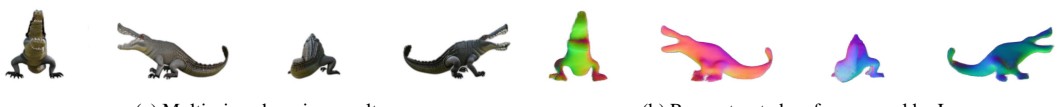

(a) Multi-view dragging results          (b) Reconstructed surface normal by Lara

Figure 11: Mesh surface normal of Lara.

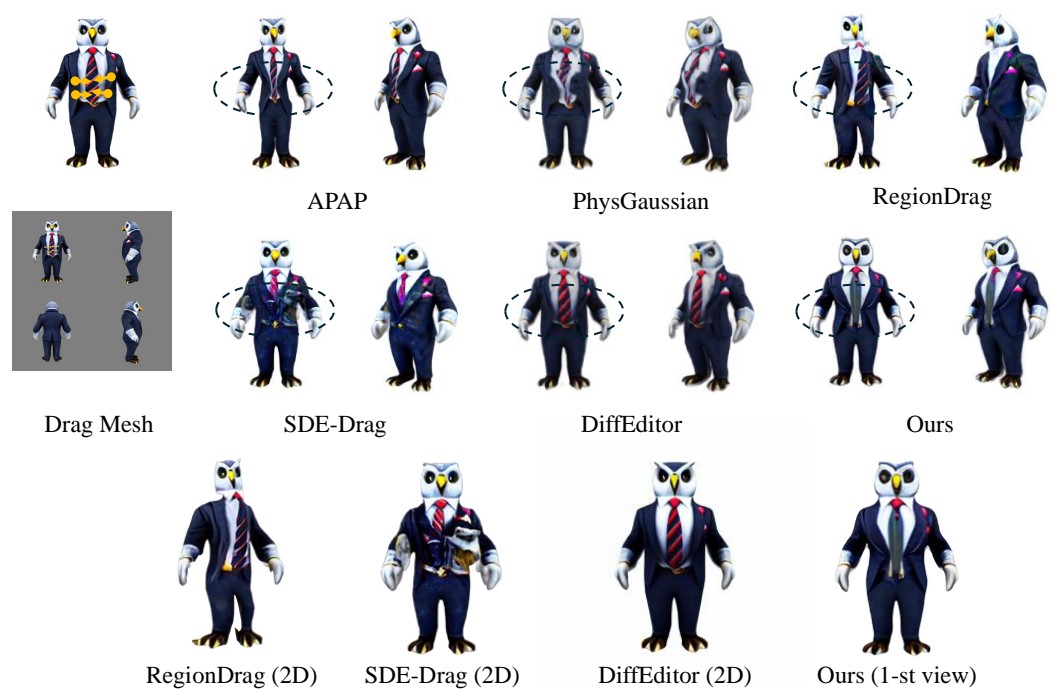

APAP     PhysGaussian     RegionDrag

Drag Mesh     SDE-Drag     DiffEditor     Ours

RegionDrag (2D)    SDE-Drag (2D)    DiffEditor (2D)    Ours (1-st view)

Figure 12: An example of failure cases. The first two rows show the 3D edited results rendered in different views. The last row visualizes the 2D editing results of three 2D baselines. In this example, our goal is to close the owl suit. Although our method successfully closes the suit to a certain degree, it still fails to reach the final position perfectly and introduces an unintended style change in the tie, as shown in the dashed circle region.

Table 3: User study on 3D Dragging results for all testing data. We calculated the proportion of results that users were most satisfied with among the comparison methods.

| Method | APAP | | PhysGaussian | | DiffEditor | | Ours | |
|---|---|---|---|---|---|---|---|---|
| | Mesh | 3DGS | Mesh | 3DGS | Mesh | 3DGS | Mesh | 3DGS |
| User preference (↑) | 20.7% | – | 5.7% | 15.1% | 13.4% | 17.3% | **60.2%** | **67.5%** |

## A.8 USER STUDY

We also conducted a user study to compare our method with others, focusing on a comprehensive assessment of editing quality, specifically how well the results match the dragging intention and exhibit the best visual quality. Participants were shown a reference image with dragging trajectories alongside all 3D editing results. The options were presented in a shuffled order, and there was no time limitation for responses. We received 62 responses to the survey. As shown in Table 3, the results demonstrate that our method outperforms others on both 3D Gaussian and mesh models, achieving the best performance in terms of user preference.

## A.9 LIMITATIONS

Firstly, the editing quality can occasionally alter the object's identity (the tie part of the owl suit in Fig. 12). The intended edit in this case is to close the suit. While our result achieves this goal to some extent, it still fails to reach the final position perfectly and introduces an unintended style change in the tie. This limitation arises because accurately adjusting the suit's position necessitates significant modifications to the tie area, where nearly half of the tie will be overlapped by the suit. Consequently, the gradient-guided editing mechanism modifies the latent noise in this region and completely relies on the diffusion prior to generate a semantically plausible result. However, this

process inherently entangles dragging accuracy (e.g., closing the suit), identity preservation (e.g., maintaining the tie's style), and global visual plausibility, making it challenging to fully satisfy all these aspects simultaneously. This issue is also common in current drag-based image editing approaches (e.g., DiffEditor and DragonDiffusion) and video editing methods (e.g., DragNUWA (Yin et al., 2023)) and remains a challenging problem to address. How to achieve more precise local control is non-trivial. Secondly, despite achieving consistent results, the four-view image editing process sometimes requires significant parameter tuning, highlighting the need for a simpler, more user-friendly multi-view editing tool, akin to InstantDrag (Shin et al., 2024). Finally, while we use multi-view images as a 3D proxy, dragging points can sometimes become occluded in all views. This limitation motivates future work on training a "pure" 3D generative model to enable more flexible and accurate 3D editing.

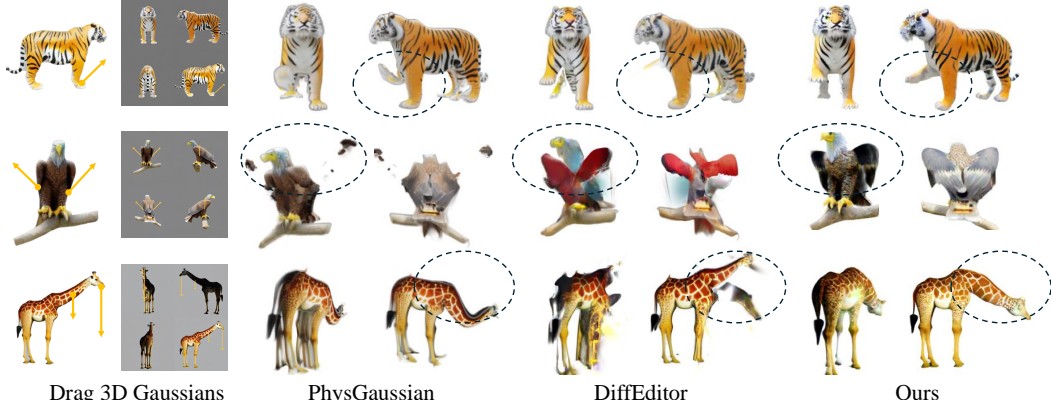

Drag 3D Gaussians      PhysGaussian      DiffEditor      Ours

Figure 13: More 3D dragging results on 3D Gaussians. Black dashed circles indicate some detailed differences. The key strength of our method lies in its ability to handle significant structural changes and generate new content during drag-based editing.

### A.10 MORE VISUAL RESULTS

Fig. 13 presents detailed qualitative results for several challenging cases. These examples demonstrate that our approach effectively handles significant structural changes and generates new content during drag-based editing. In contrast, existing baseline methods struggle to support these types of complex edits.

