# OpenReview forum: "MVDrag3D: Drag-based Creative 3D Editing via Multi-view Generation-Reconstruction Prior"
_ICLR.cc/2025/Conference — Submitted to ICLR 2025_

### Official Review · Reviewer_bkUz · 2024-10-31

**Soundness:** 3
**Presentation:** 2
**Contribution:** 2
**Rating:** 5
**Confidence:** 2

**Summary:**

The paper proposed a new algorithm called MVDrag3D, which is a framework for drag-based 3D editing. The method first generates multi-view images, followed by mechanisms introduced to enforce consistencies in the generation process, before reconstructing a 3D model. In the experiments, it looks like the proposed approach provides a more precise 3D reconstructed results comparing to other state of the art approaches

**Strengths:**

The paper is a system paper but t does has some interesting novel ideas that might be beneficial to the research community. There are several notable strengths
- The proposed approach is sensible in that leverating multi-view to control the dragging behavior is reasonable, and the overall formulation of the algorithm is intuitive.
- The introduced multi-view guidance score and the refinement approach is novel in that it helps reduce artifacts and addressed softspots in the algorithm. There are several mini-ideas in the approach that are interesting, such as adding random noises in the inversion, etc.
- Experimental results looks competitive, especially some of the visual comparisons.

**Weaknesses:**

I think there are merits in the paper but i've got some questions or confusions while reading the paper
- In figure 3, the differences of the final reconstruction results are very minor to me. I couldn't fully appreciate the need of the random noise unless I zoom into the pictures. Technically none of those results matches the input very well.
- Experimental results are conducted on vey simple 3D objects. Maybe this is a common issue for all state of the art approaches. Similarly, in fig. 6 -- many appraoches perform quite competitively (such as the boots, flower, and the suited animal case), and I struggle to tell if the approach is truly groundbreaking or incremental
- The descriptions of the technical formulations of the approach requires more details. For example, i struggled to understand the "Gaussian appearnace optimization" section -- "Note that all Gaussian properties are optimized during this process, with densification and pruning operations enabled." -- which densification and pruning approach exactly, and how is this done? The "shoe" example in fig.5 is also showing tiny improvements with the optimizations.
- It'd be good if the paper could provide human evaluations. The automatically computed metrics might not tell the full story of the effectiveness of the approach.

**Questions:**

I have raised questions in the weakness section

---

> ### Author Response · Authors · 2024-11-21
> **Official rebuttal comments to Reviewer bkUz**
>
> **Minor difference in Fig. 3**. Thank you for your detailed observation. In the revised paper, we use yellow dashed boxes to highlight the regions with evident differences. Additionally, we compute the CLIP similarity between the reconstructed images and the original image to quantitatively demonstrate the difference. Specifically, the CLIP similarity between (a) and (d) is 88.59, whereas it improves to 93.59 between (a) and (e) by the DDIM inversion with random noise. we also included a new example in Fig. 10 of the Appendix to further illustrate the advantages of DDIM inversion with random noise.
>
> Theoretically, getting an inverted noise that aligns with the Gaussian distribution is important for editing, as evidenced by numerous prior works (e.g. I2VEdit, DragDiffusion, etc.). However, the noise derived from the original DDIM inversion clearly deviates from a Gaussian distribution, as shown in Figs. 3 and 10 (b). By introducing a straightforward random noise processing step, the DDIM inversion noise becomes more consistent with a Gaussian distribution, as illustrated in Figs. 3 and 10 (c), thus leading to better editing results.
>
> **Similar results in simple cases**. Thank you for your careful comment. Yes, for simpler cases without the need for significant structural changes or the generation of new content—such as the shoe dragged to be shorter in Fig 6—both our method and the baselines (e.g. APAP) perform well. However, we would like to emphasize that the key strength of our method lies in its ability to handle significant structural changes and generate new content during drag editing, which is a very challenging task even though the 3D objects are simple. Examples include a bird opening its wings (Fig. 1), a lion opening its mouth while generating teeth (Fig. 1), a flower blooming (Figs. 6 and 9), and a leg lifting in a more articulated manner (Fig. 6).  In these cases, our method produces results that are accurate, multi-view consistent, and visually attractive. To the best of our knowledge, no existing method achieves comparable results. We will include more challenging cases.
>
> **Clarity of technical formulations**. In the section on Gaussian appearance optimization, we have revised the technical description to make it clearer and easier to understand.
>
> The “densification and pruning” process involves creating, removing, or adjusting geometry that is inaccurately positioned or sized. The pruning step removes Gaussians with opacities below a specified threshold, ensuring cleaner geometry. The densification stage is straightforward: it clones Gaussians to improve the coverage of small-scale geometry. For example, if a small-scale feature is represented by a single large splat, it is split into two smaller splats to provide a more detailed representation.
>
> Since this is a standard step in 3D Gaussian Splatting [Kerbl et al., 2023], we adhered to the implementation provided in the original paper and its released official code. As such, we chose not to elaborate further on it in the main paper.
>
> Regarding the “shoe” example in Fig. 5, we have replaced it with a new example. Additionally, we have included an ablation video in the supplementary materials, showcasing more examples to demonstrate the effectiveness of this step. Furthermore, since the evaluation of appearance is subjective to some extent, our use of GPTEval3D revealed an improvement in the Texture Details metric after SDS optimization, as shown in Table 2.
>
> **Human evaluation**. Thank you for your constructive suggestions. We have conducted a questionnaire survey as shown in the appendix (Table 3 and Sec A.8).

---

> > ### Author Response · Authors · 2024-11-26
> >
> > Dear Reviewer,
> >
> > As the rebuttal phase approaches its end with only one day remaining for PDF revisions, we would like to kindly remind you of our responses to your comments and questions.
> >
> > We have carefully addressed all your feedback and have made thoughtful updates to the manuscript to address your concerns. We hope that our efforts meet your expectations and provide clarity on the points you raised.
> >
> > If you have any remaining questions or concerns, we would be happy to discuss them further and make additional revisions as needed. Otherwise, if you find our updates satisfactory, we kindly invite you to consider reevaluating your score.
> >
> > Thank you again for your time, thoughtful feedback, and invaluable contributions to improving our work.

---

> > > ### Comment · Reviewer_bkUz · 2024-11-26
> > > **reply**
> > >
> > > I thank authors for providing clarifications and addressing my concerns. I also thank the area chairs for helping me make my questions more actionable. The revised paper is clearer and I like that the authors provided user studies for demonstrating the effectiveness of the approach (it'd be good if demographics are listed in the user study though).
> > >
> > > However, after reading the other reviewers' comments, it seems that we all have a similar concern that the experimental results are conducted on simpler datasets and visually it shows little improvements. While I appreciate authors' time addressing my questions, my rating remains unchanged.

---

> > > > ### Author Response · Authors · 2024-11-29
> > > >
> > > > Thank you for your thoughtful and detailed feedback. We appreciate your continued engagement with our work and the opportunity to address your concerns.
> > > >
> > > > **Experimental results on simpler datasets.** For the testing dataset, we would like to clarify the differences and highlight the advantages of the proposed method compared to the baselines.
> > > >
> > > >   We have included more challenging cases, as visualized in Fig. 13 in the appendix. These cases often require significant structural modifications, such as a giraffe lowering its head or an eagle spreading its wings, which involve both **large structural changes** and **the generation of new content**. For example, in the bird case (Fig. 1 or 13), the initial 3D model shows folded wings pressed against the body, resulting in shared surfaces between the wings and the body. Editing this model to represent spread wings necessitates altering the topology of the body, generating the inner wing surfaces, and reconstructing the newly exposed body regions.
> > > >
> > > >   Existing baselines, such as APAP and PhysGaussian, are restricted to deforming 3D surfaces without generating new contents or synthesizing plausible details. Similarly, 2D editing methods like DiffEditor can perform edits from a single view but fail to ensure consistency across multiple views, leading to structural inaccuracies in reconstruction.
> > > >
> > > >   Our experimental results demonstrate that, while our method may introduce certain detail artifacts (primarily due to the employed generation-reconstruction priors), it significantly outperforms the baselines in these challenging cases (as also noted by Reviewers Dkc5 and xmM9). It enables large-scale structural changes and generates new content with intuitive drag operations, making it more effective for complex scenarios.
> > > >
> > > >   Finally, we summarize the main differences and improvements between the baseline methods and our approach. For additional details, please refer to the common concern section.

---

### Official Review · Reviewer_MdMJ · 2024-11-03

**Soundness:** 3
**Presentation:** 2
**Contribution:** 2
**Rating:** 5
**Confidence:** 5

**Summary:**

This paper proposes MVDrag3D, which utilizes a deformation network and SDS loss from multi-view diffusion models to edit 3D objects. The results show that the proposed method can generate 3D creations with the corresponding drag operation while creating new textures.

**Strengths:**

* MVDrag3D is capable of user-friendly drag operations. Extending the idea of 2D drag to 3D makes sense and shows good qualitative results.
* MVDrag3D shows the ability to generate new textures, which makes it different from the standard 3D deformation works.

**Weaknesses:**

* Low-quality 3D creations: Most of the results are based on simple 3D creations or generated 3D creations, which are pretty low-quality. For example, the shoes and the fox show a blurred texture. Such evaluation dramatically limits the application of the proposed method. In addition, the edited results in the video tend to be blurred compared with the "Drag on meshes" illustration. These observations raise two concerns: (1) will this method blur the original 3D content? (2) will this method work for high-quality 3D creations?
* Failure cases of drag operations: The drag operation does not usually seem successful in the examples. The drag on the owl is not correctly optimized since the cloth doesn't reach the target position. In addition, the texture of the tie is degenerated during this process. Such a result even sees worth than the baselines like APAP. The use of SDS loss probably hurt the results to some extent.

**Questions:**

* Listed in the wakenesses.

---

> ### Author Response · Authors · 2024-11-21
> **Official rebuttal comments to Reviewer MdMJ**
>
> **Low-quality 3D creations**. Our method may slightly blur the original 3D content. This issue arises mainly due to three factors: (i) By rendering only 4 orthogonal views, some details (shape and texture) of the 3D object may be lost; (ii) The resolution of the multi-view diffusion model used (MVDream) is currently limited to 256×256. Consequently, even if high-quality images are rendered from the 3D objects, they must be downsampled to fit this resolution, leading to inevitable detail loss; and (iii) The reconstruction model (LGM) may introduce additional losses in both geometry and texture details.
>
> Fortunately, our method is highly flexible and can incorporate advancements in 3D foundational models to address the artifacts. For example, Instant3D, with its 512×512 multi-view diffusion model, or Lara and GRM, which improve geometry reconstruction, can seamlessly replace MVDream and LGM in our pipeline.  Moving forward, we plan to incorporate Lara and Instant3D into our method, which will offer stronger reconstruction priors and generation priors respectively.
>
> Finally, we would like to emphasize that the key strength of our method lies in its ability to handle significant structural changes and generate new content during drag-based editing. For example, our approach enables edits such as a bird opening its wings (Fig. 1), an animal opening its mouth (Figs. 1 and 2), or lifting its leg in a more articulated manner (Figs. 1 and 6), and a flower in bloom (Figs. 6 and 9). These types of edits are not effectively supported by existing methods.
>
> **Failure cases**. The intended edit in the owl case is to close the suit (Fig. 12 in the revised paper). While our result achieves this goal to some extent, it still fails to reach the final position perfectly and introduces an unintended style change in the tie. This limitation arises because accurately adjusting the suit’s position necessitates significant modifications to the tie area, where nearly half of the tie will be overlapped by the suit. Consequently, the gradient-guided editing mechanism modifies the latent noise in this region and completely relies on the diffusion prior to generate a semantically plausible in-distribution result. However, this process inherently entangles dragging accuracy (e.g., closing the suit), identity preservation (e.g., maintaining the tie’s style), and global visual plausibility, making it challenging to fully satisfy all these aspects simultaneously. This issue is also common in current drag-based image editing approaches (e.g., DiffEditor and DragonDiffusion) and video editing methods (e.g., DragNUWA) and remains a challenging problem to address. We discussed this limitation in the Appendix and gave a detailed analysis.

---

> > ### Author Response · Authors · 2024-11-26
> >
> > Dear Reviewer,
> >
> > As the rebuttal phase approaches its end with only one day remaining for PDF revisions, we would like to kindly remind you of our responses to your comments and questions.
> >
> > We have carefully addressed all your feedback and have made thoughtful updates to the manuscript to address your concerns. We hope that our efforts meet your expectations and provide clarity on the points you raised.
> >
> > If you have any remaining questions or concerns, we would be happy to discuss them further and make additional revisions as needed. Otherwise, if you find our updates satisfactory, we kindly invite you to consider reevaluating your score.
> >
> > Thank you again for your time, thoughtful feedback, and invaluable contributions to improving our work.

---

> > > ### Comment · Reviewer_MdMJ · 2024-11-27
> > >
> > > I appreciate the authors’ clarification and rebuttal. However, my initial concerns remain unaddressed. Specifically, the authors attribute the blurriness of the results to the four views, the diffusion model, and the reconstruction model, suggesting that better components could improve results in the future. While this is a valid point, combining and leveraging existing methods is one of the contributions of this article. Consequently, the current blurry results and the relatively weak improvement over the baseline remain a weakness. Furthermore, while the authors claim to be the first to develop an editor capable of generating new content, there appear to be prior methods with comparable effects [1].
> > >
> > > Regarding the failure cases, I agree that these are partly due to the inherent limitations of SDS, particularly the uncontrollability of results. However, contrary to the authors' claim,  it is worth noting that similar issues have been almost addressed in existing 2D drag methods [2,3]. Besides, local editing is a widely adopted strategy for managing regional changes and could serve to mitigate some of the challenges highlighted [4].
> > >
> > > Given these concerns, my rating remains unchanged at this time. However, I am open to further discussion if other reviewers have differing perspectives on these points.
> > >
> > > [1] Dong, Shaocong, et al. "Interactive3D: Create What You Want by Interactive 3D Generation." CVPR. 2024.
> > > [2] Nie, Shen, et al. "The blessing of randomness: Sde beats ode in general diffusion-based image editing." ICLR. 2024.
> > > [3] Lu, Jingyi, Xinghui Li, and Kai Han. "RegionDrag: Fast Region-Based Image Editing with Diffusion Models." ECCV. 2024.
> > > [4] Chen, Yiwen, et al. "Gaussianeditor: Swift and controllable 3d editing with Gaussian splatting." CVPR. 2024.

---

> ### Author Response · Authors · 2024-11-29
>
> Thank you for your thoughtful, detailed feedback, and valuable literature recommendations. We appreciate your continued engagement with our work and the opportunity to address your concerns.
>
> **Differences and improvements with baselines.** Thank you for your thoughtful comments. We acknowledge that, for simple local deformation cases that do not require significant structural changes or the generation of new content—such as shortening the shoe in Fig. 6—both our method and baseline approaches (e.g., APAP and PhysGaussian) perform well. Beyond these simpler cases, Fig. 13 provides additional comparisons on more challenging scenarios that highlight the limitations of the baselines and the strengths of our approach. Besides, we have summarized the main differences between the baseline methods and our proposed method. For further details, please refer to the common concern section.
>
> **Differences with Interactive3D.** We acknowledge that Interactive3D can also generate new content thanks to the SDS loss. However, a key limitation of Interactive3D is that it heavily relies on manually selecting the 3D region to be explicitly dragged, due to the lack of a strong global-aware generative prior. Let's consider opening a bird's wings (see Fig. 1 or Fig. 13), where the wings are closely tucked against its body, with most 3D Gaussian points shared between these regions. Using Interactive3D to spread the wings requires precisely selecting the Gaussian points corresponding to the wing area and separating them from the body, which is extremely challenging and is prone to artifacts. **In contrast, our approach, with a stronger generative prior offered by the multi-view diffusion model, can automatically discover where to split the wings from the body.** We will include the comparison with Interactive3D and more discussions in the final version. We will also ensure a more precise description of this comparison in the paper.
>
> **Failure cases.** First, we clarify that the failure case shown in Fig. 12 of the revised paper is not due to the SDS loss but primarily originates from incorrect multi-view editing results. To provide a detailed analysis, we included additional 2D editing methods and their corresponding 3D reconstruction results (see Fig. 12). All results were obtained using their official codes or Gradio demos.
>
>   In the last row of Fig. 12, we compare the 2D editing results of RegionDrag, SDE-Drag, DiffEditor, and our method. Both DiffEditor and our approach fail to precisely drag the suit to the target position and modify the tie’s style. SDE-Drag alters the tie’s style but introduces incorrect textures. RegionDrag requires careful selection of editing regions to avoid style changes, but distorts the overall shape and appearance. Due to these limitations in 2D editing, the corresponding reconstruction results for these methods also fail.
>
>   We believe that all methods fail due to the inherent challenge of balancing three interconnected aspects: dragging accuracy (e.g., closing the suit exactly to the tie), identity preservation (e.g., retaining the tie’s style), and global visual plausibility (e.g. avoiding distortion). Satisfying all these criteria simultaneously remains a significant hurdle.
>
>   We appreciate the suggestion to explore local editing techniques, such as those used in RegionDrag and GaussianEditor, as potential solutions. GaussianEditor allows masking Gaussian points to update only the areas requiring edits. However, this approach differs from the task we are addressing, as it primarily focuses on adding or deleting objects and altering their styles, while our work focuses on realistic deformations, structural changes, and new content generation.
>
>   Finally, we believe that region-based editing and point-based editing represent two types of techniques or tools, despite some overlap in the effects they can achieve (such as local deformation or region transformation). Adapting region-based techniques for creative 3D editing is an exciting avenue for future exploration.

---

### Official Review · Reviewer_xmM9 · 2024-11-03

**Soundness:** 3
**Presentation:** 3
**Contribution:** 2
**Rating:** 5
**Confidence:** 5

**Summary:**

This paper proposes a 3D editing method based on sparse control point dragging, applicable to both neural implicit representations and explicit mesh representation. The proposed method is based on a multi-view diffusion model and a large model for sparse view reconstruction. Specifically, the method renders the 3D representation into images from four specified viewpoints, then uses DDIM inversion to convert these images into Gaussian noise, which is regenerated using the multi-view diffusion model MVDream. During the generation process, control point movements are incorporated as a constraint to define multi-view guided energy optimization for intermediate denoising features. The generated edited images from the four viewpoints will obtain a 3D GS representation using the large pre-trained model, LGM. However, the 3D Gaussians obtained from the four views exhibit geometric deviations and texture artifacts. The former is addressed by introducing an MLP to shift the positions of Gaussian spheres, while the latter is resolved through multi-view SDS loss optimization. The paper compares the proposed method with other similar drag-based editing methods.

**Strengths:**

-- The paper has a clear description and is well-structured, providing not only the necessary details for implementing the method but also the motivation for choosing this technical approach.

-- The method proposed in this paper is reasonable and has achieved results superior to other comparative methods.

**Weaknesses:**

-- The method proposed in this paper is straightforward but lacks sufficient technical contribution. The first step generates multi-view edited images, with its optimization method and energy terms resembling those of Dragondiffusion. The main difference is merely substituting a single-view diffusion model with a multi-view one, which is a simple and direct change requiring minimal adjustments. In the second step, the LGM is used to regenerate the 3DGS model by directly utilizing an existing pre-trained model. Finally, the adjustments made through the deformation MLP and multi-view SDS loss optimization are common practices in 3D modeling and generation. Therefore, the proposed method primarily combines existing approaches without significant improvements, leaning more towards engineering rather than showcasing technical contributions.

-- Although the proposed method outperforms existing approaches in terms of deformation effects, it is still limited by the 3DGS representation, resulting in some artifacts and blurriness at the model edges in the final output. For instance, this is evident in the green leaves of the flowers, the open mouth of the crocodile, and the open mouth of the lion in the video.

**Questions:**

-- Does the proposed method ultimately produce a 3DGS representation for editing the input mesh? What if we still want a mesh-based output? While 3DGS can extract the mesh, the quality of the extracted mesh is not very good, and it is necessary to use higher-quality representations like 2DGS for better reconstruction results.

-- While the deformation MLP and multi-view SDS loss optimization do have some effectiveness in fine-tuning the final results, they seem to be suitable primarily for relatively minor issues. For more significant geometric inconsistencies and texture artifacts, it remains unclear whether they can adequately address these challenges. And could the video include results from ablation experiments related to these two aspects?

After rebuttal:
1. Though this work is a good engineering effort, I still believe it lacks sufficient technical contribution and relies heavily on the multi-view reconstruction method. As I mentioned in the previous review, it mainly combines some existing methods with some appropriate improvements, but these improvements are not substantial enough to be considered an independent technical contribution. While this combination can solve certain problems that existing methods struggle with, it does not mean that the approach is technically novel or provides enough insight.

2. Furthermore, the artifacts in the results indicate that the proposed method is too dependent on the robustness of existing methods. The authors mention that more advanced methods can be substituted, but this further suggests that the proposed approach is just a combination of existing methods, and each of these methods needs to have robust results on its own.

3. While the proposed editing method can handle some situations that existing deformation methods cannot, editing methods generally need to preserve the original representation. Although the introduction of 2DGS can further obtain meshes from the editing results, it does not fully preserve the details and topology of the original mesh. Therefore, the proposed method’s applicable editing objects might need to be reconsidered.

Taking these points into consideration along with reviews from other reviewers, I still have concerns about accepting this paper. Hence, I maintain my original score.

---

> ### Author Response · Authors · 2024-11-21
> **Official rebuttal comments to Reviewer xmM9**
>
> **Technical contribution**. Thank you for your comment and we acknowledge that the proposed method in this paper is straightforward. However, our primary goal is to tackle a challenging and under-explored task: drag-based creative 3D editing involving significant structural changes or new content generation. Existing approaches, whether based on explicit spatial transformations or implicit latent optimization within limited-capacity 3D generative models, struggle to make substantial topology changes or generate new content across diverse object categories. For instance, APAP is unable to open the bird’s wing or the lion’s mouth (Fig. 1), PhysGaussian fails to generate complementary new content (Figs. 1 and 6), and DiffEditor struggles to achieve view-consistent 3D edits (Figs. 1 and 6).
>
> To tackle this, we reformulated drag-based 3D editing as a multi-view editing and reconstruction problem. While we draw inspiration from image editing techniques, our approach is both innovative (as noted by Reviewers Dkc5 and bkUz) and effective (validated by all reviewers) for the 3D dragging task. Furthermore, we have proposed practical solutions to each sub-problem inherent in this framework, enabling our method to achieve state-of-the-art results.
>
> We would like to emphasize that while the techniques we adopted may not be groundbreaking individually, each design was thoughtfully tailored to address a specific challenge. The effectiveness of our overall framework and the results it has achieved highlight the significance of our contributions.
>
> **Certain artifacts of the final output**. Yes, the 3D GS generated by our method still has some limitations in detail quality. This issue arises mainly due to three factors: (i) By rendering only 4 orthogonal views, some details (shape and texture) of the 3D object may be lost; (ii) The resolution of the multi-view diffusion model used (MVDream) is currently limited to 256×256. Consequently, even if high-quality images are rendered from the 3D objects, they must be downsampled to fit this resolution, leading to inevitable detail loss; and (iii) The reconstruction model (LGM) may introduce additional losses in both geometry and texture details.
>
> Fortunately, our method is highly flexible and can incorporate advancements in 3D foundational models to address the artifacts. For example, Instant3D, with its 512×512 multi-view diffusion model, or Lara and GRM, which improve geometry reconstruction, can seamlessly replace MVDream and LGM in our pipeline.  Moving forward, we plan to incorporate Lara and Instant3D into our method, which will offer stronger reconstruction priors and generation priors respectively.
>
> Finally, we would like to emphasize that the key strength of our method lies in its ability to handle significant structural changes and generate new content during drag-based editing. For example, our approach enables edits such as a bird opening its wings (Fig. 1), an animal opening its mouth (Figs. 1 and 2), or lifting its leg in a more articulated manner (Figs. 1 and 6), and a flower in bloom (Figs. 6 and 9). These types of edits are not effectively supported by existing methods.
>
> **Final output when editing a mesh**. The final output of our method is a 3D GS representation for 3D meshes. If a mesh model is still required, it needs to be extracted from the 3D GS, which may result in some loss of detail. However, our primary focus is on handling significant structural changes and generating new content during drag-based editing. As noted in the comment, 2D GS could be a potential solution for improved mesh extraction. Additionally, we found that the open-source work Lara (ECCV 2024) could leverage four views to feed-forwardly regress a 2D GS model with a smoother surface, as illustrated in Fig. 11 of the Appendix. In the future, we plan to release a Lara-based version of our method.
>
> **Effect of deformation MLP and multi-view SDS loss optimization**. Thank you for your thoughtful comment. In fact, we have observed that LGM is capable of reconstructing relatively good results. While there is some degree of texture loss and geometric misalignment, there are no significant geometric inconsistencies or noticeable texture artifacts. The worst case we have encountered was the crocodile example in Fig. 2, where the tail is misaligned. Here, our solution can correct this effectively. Additionally, we have provided a revised video demo in the supplementary material, showcasing additional results for the ablation study of the two optimization stages. This demonstrates the individual contributions of the deformation MLP and the multi-view SDS loss optimization.

---

> > ### Author Response · Authors · 2024-11-26
> >
> > Dear Reviewer,
> >
> > As the rebuttal phase approaches its end with only one day remaining for PDF revisions, we would like to kindly remind you of our responses to your comments and questions.
> >
> > We have carefully addressed all your feedback and have made thoughtful updates to the manuscript to address your concerns. We hope that our efforts meet your expectations and provide clarity on the points you raised.
> >
> > If you have any remaining questions or concerns, we would be happy to discuss them further and make additional revisions as needed. Otherwise, if you find our updates satisfactory, we kindly invite you to consider reevaluating your score.
> >
> > Thank you again for your time, thoughtful feedback, and invaluable contributions to improving our work.

---

### Official Review · Reviewer_Dkc5 · 2024-11-03

**Soundness:** 3
**Presentation:** 3
**Contribution:** 3
**Rating:** 8
**Confidence:** 5

**Summary:**

In this paper, the authors present a framework named MVDrag3D for drag-based 3D editing. It first renders a 3D object and the drag points into 4 orthogonal views, and introduce a multi-view guidance energy to achieve consistent multi-view score-based image editing. Multi-view Gaussian reconstruction is then performed on the edited images, followed by Gaussian position adjustment by view-specific lightweight deformation networks. Finally, an image-conditioned multi-view SDS optimization is applied to further enhance view consistency and visual quality.

**Strengths:**

+ The idea of casting 3D drag-based editing into multi-view 2D drag-based editing sounds novel and feasible.
+ The use of multi-vew diffusion model to ensure consistent multi-view image editing sounds novel and feasible.
+ The lighweight view-specific deformation networks appear to be effective in improving the geometric alignment of the 3D Gaussians with the image.
+ The use of image-conditioned multi-view SDS optimzation for enhancinmg view consistency and visual quality sounds logical.
+ Both qualitative and quantitative results demonstrate SOTA results.

**Weaknesses:**

- There is no disucssions on how to choose the optimal 4 orthogonal views. Theoretically, the 4 orthogonal should be chosen such that the drag directions should be as far away from the view directions as possible. In this paper, the authors simply choose orthogonal azimuths (0 deg, 90 deg, 180 deg, 270 deg) and a fixed elevation (0 deg).
- By rendering only 4 orthogonal views, some details (shape and texture) of the 3D object may be lost.
- In 3D Gaussian reconstruction step, partial 3D Gaussians are regressed for each view, which are then fused into a unifed representation. It is not clear why the authors do not perform multi-view Gaussian reconstruction instead.
- It is also not clear wheather the Gaussian position optimization step is necessary. Why can't the image-conditioned multi-view SDS optimization be carried out directly on the initial 3D Gaussians?

**Questions:**

- Why do the authors regress a partial 3D Gaussians for each view and fuse them afterwards instead of performing a multi-view reconstruction?
- Referring to Table 1, is there any explanations why SDS optimization produces better results for Gaussians than for meshes in terms of DAI?
- For 3D meshes, are the final outputs still 3D Gaussians? Are the quantitative results all computed on 3D Gaussians?

---

> ### Author Response · Authors · 2024-11-21
> **Official rebuttal comments to Reviewer Dkc5**
>
> **View selection**. The current four view choice is primarily motivated by two reasons: (1) These four views can already express the 3D drag trajectory relatively completely. Even in cases where the drag direction aligns with one of the four views, it will still be visible from another side view unless complex occlusion occurs. Our experiments have demonstrated that these four views perform well in most cases. (2) The multi-view generation and reconstruction models we employed exhibit a strong bias toward these specific views. For example, if we use the other four fixed orthogonal views, we observe a significant degradation in the reconstruction quality of LGM.
>
> We also acknowledge that relying solely on orthogonal views may not be optimal and this highlights an intriguing future direction: exploring dragging on native 3D generation models (e.g., CLAY) or video-like multi-view diffusion models, like SV3D, which supports 21 views.
>
> **Some details lost by only 4-view rendering**. For a more detailed discussion, please refer to our response to Weakness 2 from Reviewer xmM9.
>
> **Multi-view reconstruction**.  We understand that the term "multi-view reconstruction" in the comment refers to the traditional method of 3D reconstruction using multiple posed input images. We agree that this approach is theoretically possible. However, our setting, actually, presents a sparse-view 3D reconstruction problem, using only four views. Traditional reconstruction methods typically require at least dozens of posed views to achieve satisfactory results. Although recent advancements, such as ReconFusion, have shown promise, they often face challenges like quality degradation when the number of input views is rather limited.
>
> In contrast, a key observation in our work is that pre-trained feed-forward multi-view reconstruction models (e.g., LGM, GRM, and Lara) have embedded strong geometric priors for object reconstruction and can offer significant advantages, including generally plausible quality, fast inference (in seconds), and ease of application. Therefore, we directly leverage these multi-view reconstruction models in our approach.
>
> **Directly applying image-conditioned SDS optimization**. Yes, directly performing image-conditioned SDS optimization can work to some extent. However, this process often results in less visually pleasing results (see the new video demo). We analyzed the main reasons as follows: The 3D GS produced by LGM is under-reconstructed, exhibiting a certain degree of geometric misalignment and texture loss. The inherent ambiguity of the SDS loss makes it challenging to achieve an optimal result when simultaneously optimizing both the texture and geometry of the 3D GS, which has an excessive degree of freedom for optimization. To address these issues, we introduced a deformation network that serves as an effective regularization to eliminate low-frequency geometric mismatches. With this better-aligned 3D GS, we then apply SDS to fine-tune its appearance. This divide-and-conquer strategy leads to improved visual effects.
>
> **SDS produces better DAI on 3DGS than on meshes**.  The DAI value depends on the accuracy of multi-view image editing and the quality of 3D reconstruction (including initial reconstruction and two optimization stages). Regarding why SDS produces better DAI on 3D GS than on meshes, we analyze the reasons as follows:
>
> Firstly, for the accuracy of multi-view image editing, the 3D GS models we selected are generated using LGM, which relies on MVDream first to generate four images and then reconstruct the 3D GS accordingly. As a result, the image distribution of LGM aligns more closely with the training distribution of MVDream, giving better DDIM inversion and more accurate multi-view editing. In contrast, the renderings of mesh models, even with the use of certain techniques, cannot ensure perfect inversion, causing some detail loss during editing (see Figs. 3 and 10). Secondly, during the SDS-based appearance optimization, we compute the LPIPS loss using the four edited images. This process may propagate errors originating from these edited images. These combined factors ultimately impact the DAI value, which is worse for mesh models compared to 3D Gaussian models.
>
> Lastly, as analyzed above, we note that SDS-based appearance optimization cannot perfectly guarantee dragging accuracy. However, we incorporate this design to enhance the visual quality with sharper and clearer details. To illustrate the differences between using SDS and not using SDS, we have included additional video results in the revised video demo. This observation is further supported by the Texture Details metric evaluated using GPTEval3D (see Table 2 in the revised paper).
>
> **Final output when inputting mesh**. The final output of our method is 3D GS for 3D meshes. To ensure a fair evaluation, we compute all quantitative results on the images rendered from these outputs (either 3D GS or meshes).

---

> > ### Comment · Reviewer_Dkc5 · 2024-11-25
> >
> > Thank the authors for providing their feedback which addresses my concerns. I am happy keep my initial rating.

---

> > > ### Author Response · Authors · 2024-11-25
> > >
> > > Thank you for your thoughtful and insightful review and for confirming that our feedback has addressed your concerns. We truly appreciate your time and effort in evaluating our work.

---

### Author Response · Authors · 2024-11-25

Dear Reviewers,

Thank you for your valuable feedback and insightful comments.

As the discussion period concludes on November 26th, we kindly ask if our responses have effectively addressed your questions. Please don’t hesitate to reach out with any additional questions, concerns, or requests for clarification.

Warm regards, The Authors

---

### Author Response · Authors · 2024-11-29
**Response to common concerns of Reviewer bkUz and MdMJ**

**Differences and improvements with baselines.** To better highlight the differences and improvements between the methods, we categorized drag editing tasks into three types: **local deformation**, **large structural changes**, and **new content generation**. We analyzed and compared the results of four methods—APAP, PhysGaussian, DiffEditor, and Ours—across these categories, as summarized in the table below.

| Editing Category | Local Deformation  | Large Structural Changes | New Content Generation |
| :---- | :---- | :---- | :---- |
| Examples | *Yoda* in Fig. 1, *black shoe* and *axes* in Fig. 6 | *Dog* in Fig. 1, *wolf*, *elephant*, *chair* in Fig. 6, *tiger* and *giraffe* in Fig. 13 | *Sparrow* and *lion* in Fig.1, *crocodile* in Fig. 2,  *lotus bud* in Fig. 6, and *eagle* in Fig. 13 |
| APAP | √ | less realistic shape | × |
| PhysGaussian | √ | less realistic shape | × |
| DiffEditor | Inconsistent | Inconsistent | Inconsistent |
| Ours | √ | √ | √ |

For **local deformation** tasks, such as deforming *Yoda* in Fig. 1 or shortening black shoes and stretching axes in Fig. 6, APAP, PhysGaussian, and our method handle these tasks successfully, while DiffEditor exhibits inconsistencies.

For **large structural changes**, such as lifting the leg of the *dog* in Fig. 1 or articulating the *wolf* and *elephant* in Fig. 6 and the *tiger* and *giraffe* in Fig. 13, APAP and PhysGaussian produce unrealistic shapes. For instance, when lifting the wolf’s left leg in Fig. 6, APAP deforms the leg but bends it rather than lifting it, resulting in an unrealistic shape. In contrast, our method produces realistic and natural articulation-like shapes that adhere to the animal’s kinematics.

For **new content generation**, such as opening the *lion*’s mouth in Fig. 1 or opening the *eagle*’s wings in Fig. 13, APAP and PhysGaussian are unable to handle these tasks. While DiffEditor can generate new content from certain views, inconsistencies across multiple views result in degraded reconstruction quality. In contrast, our method handles these cases effectively.

Finally, as summarized in the table, our method outperforms the baselines in most cases (as also noted by Reviewers Dkc5 and xmM9). It enables large-scale structural changes and generates new content through intuitive drag operations, making it highly effective for complex scenarios. We believe the main contribution of this work lies in a new framework to address a very challenging and underexplored task, while the minor artifacts could be addressed by adopting more powerful generation-reconstruction priors in the follow-up works.

---

### Meta-Review · Area_Chair_wzdp · 2024-12-21

**Metareview:**

The paper proposes a method for 3D drag editing. Most reviewers gave negative ratings, with the primary concern being the low visual quality and marginal qualitative improvement in the results. The rebuttal addressed the paper’s ability to handle significant structural changes, but it did not sufficiently convince the reviewers that the improvements were substantial enough to justify the method. Despite Reviewer Dkc5 recommending acceptance, he/she did not feel strongly enough to champion the paper, and the results still appeared blurry. After a careful review of the results, the area chair appreciates the significant quantitative improvement (especially in terms of DAI) and recognizes the merit of the work. However, the visual quality remains an issue. Therefore, the area chair recommends rejection at this stage, and suggests the authors to revise the paper by incorporating the suggestions from the reviewers and submit the revised version to a future venue.

**Additional Comments On Reviewer Discussion:**

Most reviewers gave negative ratings, with the main/common concern being the low visual quality and marginal improvement of the results. The rebuttal emphasized the ability to handle significant structural changes and generate new content during drag-based editing; however, it did not sufficiently convince the reviewers. While the area chair appreciates the significant quantitative performance in DAI, the qualitative results still appear blurry. While Reviewer Dkc5 recommends acceptance, he/she did not feel strongly enough to champion the paper.

---

### Decision · Program_Chairs · 2025-01-22

Reject